# The Effects of Horticultural Therapy on Sense of Coherence among Residents of Long-Term Care Facilities: A Quasi Experimental Design

**DOI:** 10.3390/ijerph19095412

**Published:** 2022-04-29

**Authors:** Ruo-Nan Jueng, I-Ju Chen

**Affiliations:** 1Department of Nursing, National Yang Ming Chiao Tung University Hospital, Yilan 260006, Taiwan; 2Department of Nursing, School of Nursing, National Yang Ming Chiao Tung University, Taipei 112304, Taiwan

**Keywords:** horticultural therapy, sense of coherence (SOC), older adults, long-term care facilities (LTCFs)

## Abstract

Promoting positive mental health is crucial for the elderly living in long-term care facilities (LTCFs). This study aims to examine the effectiveness of horticultural therapy on the level of sense of coherence (SOC) among older LTCF residents with relatively normal mental function. With convenient sampling, a total of 86 participants were recruited from 12 LTCFs in northeastern Taiwan. In the experimental group (*n* = 49), the mean (±standard deviation) score of SOC was 50.45 ± 6.07 at baseline and increased to 56.37 ± 7.20 (*p* < 0.001) after 12-week horticultural intervention. In contrast, the mean SOC score did not change significantly in the control group (*n* = 37) during the study period. Generalized estimating equation analysis showed that a significant interaction effect between group and time on the SOC score (*p* < 0.001). Our findings indicate that horticultural therapy is effective to strengthen the SOC level of older LTCF residents without dementia.

## 1. Introduction

Population aging has become a rising concern in many countries. Aged people have greater health problems and long-term care needs than younger people, leading to increased health care expenditure [1]. Long-term care facilities (LTCFs) are often the places where people with complex health needs that cannot be met in a community setting reside and are cared for until their death [2]. An increase in the provision of institutional care services has been observed [3].

The growing number of frail older adults and vulnerable persons with disabilities necessitate a multidisciplinary response from across the care continuum [4]. In addition to limited physical function, frail older adults are at high risk for mental disorders such as depression and loneliness, especially those living in LTCFs [5]. Unlike living at home, older residents in LTCFs cannot always be accompanied by family, and the environments of many LTCFs are usually monotonous, which may be detrimental to their mental health. The World Health Organization stated that mental health is “a state of complete physical, mental, and social well-being and not merely the absence of disease or infirmity” [6]. Thus, interventions that aim to enhance the positive mental health of frail or prefrail older residents of LTCFs should be investigated.

Antonovsky first proposed the concept of sense of coherence (SOC), which refers to individuals’ subjective perceptions of their mental health, rather than medical diagnoses of mental illness [7]. In salutogenic theory, which explains the origins of health and describes how health can be improved, the SOC is the core concept and focuses on the process by which individuals use environmental resources to face challenges in daily life and effectively cope with their stress and feelings [8]. SOC includes three core components: comprehensibility, manageability, and meaningfulness. Comprehensibility is the degree to which events are perceived to be explicable, predictable, and structured. Manageability is the degree to which one feels they can cope. Meaningfulness is how much one feels that life makes sense, and how worthy challenges are of investment and engagement. Thus, SOC can be understood to represent an individual’s ability to maintain a positive attitude while exhibiting understanding amidst challenging situations (comprehensibility), applying diverse resources (manageability), and seeking and realizing meaning in life (meaningfulness), all of which can help maintain an ideal state of health [9,10,11,12]. Relevant studies have proven that SOC can be a crucial indicator of the mental health of older adults [13,14]. Enhancing SOC can promote mental health.

Our previous study showed that older adult residents of LTCFs in northeastern Taiwan had relatively low SOC scores and their SOC status could be influenced by education level, activities of daily living, environmental factors, and interactions between personal and environmental factors [12]. The importance of the balance between environmental demands and individual competence in old age has been addressed by Lawton’s ecological model of aging, which emphasizes the psychological benefits from the appropriate balance [15]. It has been reported that older adults in LTCFs can enhance their physical and mental health by increasing their interactions with others among the diverse activities [16].

Horticultural therapy is defined as horticulture-related activity by individuals with the aid of a facilitator to improve diverse outcomes [17]. As one of the nonpharmacological interventions, horticultural therapy has received increasing attention from researchers in recent decades [18]. Positive psychological, social, and physical health benefits of horticultural therapy such as improvement in memory, cognitive abilities, sense of hope, task initiation, language skills, socialization, muscle strength, balance, endurance, and coordination have been documented in the recent literature [18,19,20,21,22]. A meta-analysis study indicated that the use of participatory horticultural therapy has been proven to be effective in promoting cognitive function, agitation, positive emotion, and engagement of people with dementia [18]. In addition, it has been systemically determined that physical functioning (upper body flexibility and aerobic endurance) in older adults with cancer and the psychological outcomes (emotional functioning and well-being, subjective social functioning, and quality of life) in the elderly can by improved with the intervention of horticultural therapy [21].

SOC, developed during childhood and early adulthood, was considered to become stable after the age of 30 years [7]. However, several studies have demonstrated that it is possible to strengthen SOC levels with some interventions, even in adulthood [23]. So far, the effectiveness of horticultural therapy for improving the SOC status of older residents in LTCFs is still poorly understood. According to a review of relevant literature, we hypothesize that the elderly in LTCFs would benefit from participating in the program of horticultural therapy in terms of improvement in their SOC status. This study aimed to investigate the effectiveness of horticultural therapy on the scores of SOC among older LTCF residents without clinically significant dementia in northeastern Taiwan.

## 2. Materials and Methods

### 2.1. Study Design and Participants

This study employed a quasi-experimental research design. With convenient sampling, 12 out of 39 LTCFs registered in Yilan County in 2015 were recruited. These 12 LTCFs were randomly assigned to the experimental (*n* = 6) and control (*n* = 6) facilities. To minimize contamination at the participant level, the eligible residents in the six experimental LTCFs were assigned into the experimental group, and those in the six control LTCFs into the control group. The inclusion criteria were as follows: residence in the LTCFs for at least three months, aged 65 years or above, clear consciousness with an ability to communicate and express their opinions, and provided consent to participate in this study. Those who had received horticultural activities within one month before enrolment were excluded from this study. The flowchart for participant recruitment is presented in Figure 1. Participant recruitment began after this study was approved by the Institutional Review Board of National Yang Ming Chiao Tung University Hospital (IRB No. 2016B006). Data were collected from 25 June to 31 December 2016.

### 2.2. Horticultural Activities Design Basis

The horticultural activities were designed based on the characteristics of the aged living in the LTCFs including their sleep–wake rhythm, cognitive function and limited physical capabilities, physical environment of LTCFs, and plant species that should be easy to obtain and grow, and are inexpensive. Therefore, indoor desktop gardening was adopted as the horticultural intervention in the present study. No outdoor activity was performed in this study in order to prevent program interruption from bad weather conditions. The plants used in the activities included sweet potato, mung bean, evergreen, sweet potato, bamboo cypress, bamboo cypress, broad-leaved Podocarpus, tea, rose, green beans, common jasmine orange, sprouts, and succulent arrangement. The wide range of plants for cultivation or ornament aimed to stimulate multiple senses and creativity of the aged. A 40-min session was held once per week for 12 weeks. According to the LTCFs residents’ routine schedules, these sessions were held at 10:30–11:30 a.m. or 2:30–3:30 p.m. To increase interpersonal interaction, all the sessions were practiced in the form of small group activities. With the aid of activity practitioners and teamwork, participants may feel empowered and experience a sense of accomplishment by managing to fulfill plant cultivation. Moreover, participants may have opportunities to reflect the meaning of life by watching plants grow over time. With both empowerment and reflection processes in the horticultural program, we aimed to strengthen the SOC status of participants in the experiment group. In contrast, the established traditional activities were continued for the control group during the research period.

### 2.3. Measures

A pretest was conducted one week before the intervention, and post-tests were administered at the 4th and 8th weeks of the research period as well as one week after the intervention was completed (the 12th week). The structured questionnaire used in the present study comprised four parts: personal factors, physical-environment factors, social-environment factors, and the SOC scale. Personal data were demographic data (age, sex, marital status, education level, and religion) and health status (number of multi-morbidities, score of Barthel index for activities of daily living (ADL_S_) [24], score of Lawton instrumental activities of daily living (IADL_S_) scale [25], mini-mental status examination (MMSE) score [26], and geriatric depression scale (GDS)-15 score [27]). Physical-environment factors covered room type, the presence of natural window views, and the presence of outdoor public spaces. Social-environment factors were number of leisure activities in LTCFs per week, number of family visits per week, and number of LTCF staff (including full-time registered nurses, nurse aides, and social workers).

This study adopted the Chinese version of Antonovsky’s short 13-item SOC scale. The Chinese SOC scale was validated by Tang and Dixon (Cronbach’s α = 0.89) [28]. The short-form SOC scale is nearly equivalent to the long-form SOC scale in reliability and validity [8]. The short-form SOC scale features five items for comprehensibility, four for manageability, and four for meaningfulness. Each item was rated on a 7-point Likert scale from 1 (never) to 7 (very often) and the total score ranged from 13 to 91. Higher scores indicate stronger SOC.

### 2.4. Data Analysis

Data analyses were performed using SPSS 20.0, and *p* < 0.05 was set as the level of significance. All data were expressed as a percentage or mean ± standard deviation (SD). The differences between the two groups in terms of personal factors, physical-environment factors, social-environment factors, and the SOC scores were determined by the independent *t* test or Pearson’s χ^2^ test as appropriate. The differences in health outcomes before and after intervention within the same group were determined with the paired *t* test. Pearson’s correlation analysis was used to explore the relationship between SOC and other mental health outcomes. The repeated measures analysis and the data dependency issue was dealt with the general estimating equation (GEE) with the first-order autoregressive error structure. The independent variables in the multivariate GEE models included group, time points, and interaction term between group and time point. All the baseline characteristics, except SOC score, were adjusted as covariates in the GEE model.

## 3. Results

### 3.1. Baseline Data

A total of 86 participants were finally recruited into the study. There was no significant difference between the experimental and the control groups in terms of demographic characteristics, health factors, environmental factors, and SOC scores at baseline (Table 1). Among all 86 participants, the average age was 81.76 ± 8.47 years, and the majority of the participants were female (63.96%), divorced or widowed (72.10%), had religion (84.79%), and illiterate (50.0%). Regarding health factors, 80.24% had two or more chronic diseases, 55.83% with mild to moderate physical disabilities (ADLs score ≥ 41), 84.89% with normal cognitive function (MMSE score ≥ 24), and 79.07% without depressive symptoms (GDS score < 5). The mean ADLs, MMSE, and GDS scores of all participants were 60.20 ± 3.42, 23.7 ± 1.4, and 4.60 ± 3.51, respectively. In terms of the environmental factors in the LTCFs, 86.5% lived in the room with roommate(s), 60.47% without a natural window view, and 75.59% without outdoor activity space. The average staff number in the LTCFs was 15.53 ± 6.78). The majority of residents (89.79%) were visited by their family members or friends once a week, and 73.26% had one to two leisure activities per week in the LTCFs.

Before the intervention of horticultural therapy, the mean SOC score was 50.45 ± 6.07 (range: 37–61) in the experimental group and 52.97 ± 6.00; (range: 38–63) in the control group, which was not significantly different (*p* = 0.059). There was no significant relationship between SOC and MMSE (r = 0.009, *p* = 0.929) and GDS (r = −0.066, *p* = 0.507) at the baseline.

### 3.2. The Change of SOC Scores before and after Intervention in the Experimental Group

At baseline, the mean SOC score of the experimental group was 50.45 ± 6.07. After initiation of horticultural therapy, there was a significantly increasing trend in the mean SOC score of the experimental group (Table 2). The mean score measured at the 4th and 8th week after intervention was 51.06 ± 6.28 and 56.22 ± 7.19, respectively. The improvement in SOC score was maintained at the 12th week (56.37 ± 7.20). Even with four more sessions, the mean SOC score at the 12th week was close to that at the 8th week. In contrast, there was no significant change in MMSE (*p* = 0.273) and GDS (*p* = 0.112) between the baseline and 12th week.

### 3.3. Comparison of SOC Score between the Experimental and Control Groups

The mean SOC scores of the experimental and control groups at various time points are displayed in Table 3. There was no significant difference in mean SOC score between two groups at baseline and at the 4th week after intervention. However, after 8-week intervention, the experiment group had a significant higher mean SOC score (56.22 ± 7.19) than the control group (53.22 ± 5.96). The difference between the two groups still persisted at the 12th week.

### 3.4. Difference of SOC Score Improvement between the Experimental and Control Groups at Various Time Points

Table 4 shows the result of multivariate GEE analysis. After the time and group factors were considered, the time points (i.e., baseline, 4th, 8th, and 12th weeks) and groups (i.e., the experimental and control groups) were used to create an interaction term. Baseline data of the control group were used as the reference for comparing how the groups differed in the interaction with time. After adjusting for personal factors, physical-environment factors, and social-environment factors, the effect of the interaction term on SOC score improvement gradually strengthened over time (B = 0.53 at the 4th week, 5.79 at 8th week, and 5.96 at the 12th week), which suggested that the horticulture therapy enabled the participants to increase their SOC scores continuously over 12 weeks.

## 4. Discussion

In the present study, we developed a 12-week program of horticultural activities related to indoor desktop gardening for older LTCF residents with physical disabilities, and examined the effectiveness of this horticultural program on SOC score with quasi-experiment design among 86 older residents of 12 LTCFs in northeastern Taiwan. Significant improvement in SOC score over time was observed in the experiment group. This finding may provide preliminary support that horticultural activities can be a health promotion strategy to strengthen SOC in the context of LTCFs.

Based on the salutogenic model, which emphasizes the origin of health rather than disease, it has been suggested that both empowering and reflection processes should be included in the health promotion activities that aim at improving SOC level [23]. These two processes address the behavioral and the perceptual mechanisms in the salutogenic model. The behavioral mechanism highlights empowering people to use their resources in stressful situations. The perceptual mechanism is focused on encouraging people to reflect on their understanding of the stressful situation and the available resources and to feel meaningful in coping with stressors [23]. In the present study, the participants in the experimental group learned the knowledge and skills of cultivating indoor plants and put these into practice with the help of the researchers and other participants. This empowering process would make the participants feel confident and accomplished. Furthermore, the participants may have opportunities to reflect on the meaning of life by watching plants grow over time. Moreover, as a nostalgic reminder, cultivating plants helped the participants recall and reflect on their life experience regarding farming or gardening and beyond. It is presumed that the components of empowerment and reflection in the horticultural therapy improved the SOC status of the participants in the experimental group. However, the exact mechanism of how horticultural therapy can strengthen SOC is still not elucidated and needs further investigation.

Besides the SOC enhancement in the present study, according to the results of randomized clinical trials (RCTs) with horticultural activity program as intervention and mood tests as outcome measures in the context of LTCFs, the horticultural intervention could improve other aspects of positive mental health among older LTCFs residents without clinically significant cognitive impairment [22,29]. The trial being conducted by Lai et al. in Hong Kong revealed significant improvement in subjective happiness over time among older adults living in LTCFs who received a weekly 60-min horticultural program for eight weeks (*n* = 46) when compared with the controls (*n* = 50) [29]. The study being conducted by Chu et al. in southern Taiwan showed that positive attitude toward aging and enhanced sense of hope could be significantly induced in the 8-week horticultural intervention group (*n* = 45) when compared with the control group (*n* = 43) [22]. With the presence of positive emotion and good mental function, people who had high positive mental health were less likely to develop mental disorders such as depression. Given that older adults in LTCFs are at great risk of mental health problems, it is worth paying more attention to increasing positive mental health in this population.

Interestingly, the evidence of the impact of horticultural therapy on GDS score is not consistent across the studies performed among older LTCF residents with no or mild cognitive impairment [29,30,31,32,33]. Significant reduction in GDS score in the experimental group was observed in the RCT by Chu et al. (*n* = 75) and in the quasi-experimental studies by Masuya et al. (*n* = 9) and by Chen et al. (one-group, pretest–posttest design, *n* = 10) [30,31,32], but GDS score in the experimental group did not significantly change in the RCT by Lai et al. (*n* = 46) and quasi-experimental study by Park et al. (*n* = 24), which was similar to the result of our study [29,33]. The conflicting evidence may result from the methodological heterogeneity such as different type of LTFCs and lack of standardized measures. Another possible explanation for nonsignificant change in GDS score after horticultural therapy is the ceiling effect. The majority (around 80%) of both groups in our study had been within a normal range of MMSE (≥24) and GDS (<5) before intervention. The main mental benefit of horticultural therapy for those older LTCF residents who do not have cognitive dysfunction and depression symptoms may be the improvement in their positive mental health, which may be helpful to minimize the possibility of mental illness in the future.

Though the exact mechanism of how horticultural therapy can promote health is still unclear, a growing body of evidence shows that the benefits of horticultural therapy, even including activities of viewing plants, may be associated with the relief of psychological stress [34,35,36]. A pilot study revealed that when compared to other occupational therapies, horticultural therapy may modulate stress in veterans, as evidenced by decreased cortisol levels and depressive symptoms [34]. A recent study evaluating the physiological and psychological responses during the computer task demonstrated that mental stress is reduced with the presence of nearby indoor plants [35]. Furthermore, older participants of gardening activities were found to have increased levels of brain-derived neurotrophic factor, which is an important neuroprotective agent [36].

There is a wide range of activity protocol of horticultural intervention in terms of activity type, duration (overall intervention period and time length of single session), frequency, and settings (group/individual and indoor/outdoor). To meet the needs of participants with different function status, the mode of horticulture activities can be adjusted to suit their capabilities. Though the optimal protocol is still unknown, it has been reported that the overall duration of the intervention delivered seemed to have a greater impact on treatment effectiveness than the duration of each single session [37]. Based on the results of the RCT conducted in the LTCFs, activity programs with the intervention duration up to eight weeks have been proven to be effective in improving the mental health outcomes [22,29]. Most relevant studies also had horticultural activity programs lasting for 6–8 weeks. In the present study, a program with longer duration (for consecutive 12 weeks) was adopted because the main outcome measure was SOC, which was once considered a stable entity in adulthood. We intended to use a longer activity program to induce a larger effect size of SOC change among older LTCF residents. However, our results showed that the mean SOC score in the experimental group at the 8th week was almost identical to that at the 12th week, though the SOC continuously improved through the entire intervention period. From a cost-effectiveness perspective, the present study provides preliminary evidence that the 8-week program would be an option with acceptable treatment effectiveness for strengthening the SOC level of the older adults living in the LTCFs.

Several limitations in the present study should be addressed. First, the follow-up period was limited. The long-term effect of horticultural therapy on SOC remains elusive. Second, only a limited number of mental health outcomes were measured including SOC, MMSE, and GDS. Although SOC was not found to be associated with MMSE and GDS scores, further studies are needed to investigate the relationship between SOC change and other physical, psychological, and social outcomes among those receiving horticultural therapy. Third, the horticultural program in the present study did not include any outdoor activity. The comparison of the effects on SOC between indoor and outdoor activities warrants further investigation. Finally, all participants in the present study were recruited from 12 LTCFs that were sampled in northeastern Taiwan based on convenience. The generalizability of our results in other populations or other contexts is a concern.

## 5. Conclusions

The present quasi-experimental study provides preliminary evidence about the effectiveness of a 12-week horticultural activity program on strengthening SOC level among older LTCF residents without clinically significant dementia. Our findings indicate that the mean SOC score in this population was not high but could be improved by horticultural intervention over time. SOC is one of indicators of positive mental health. Even with some research limitations, the results of the present study suggest that LTCF staff can incorporate horticultural activities into part of the routine schedule of LTCFs to maintain or enhance the SOC level of their residents, which provides a reference for positive mental health improvement in older people living in LTCFs. Further studies with larger sample size, longer follow-up period, or randomized trial design are needed to investigate how to maintain or increase the effectiveness of horticultural therapy on SOC in a cost-effective way.

## Figures and Tables

**Figure 1 ijerph-19-05412-f001:**
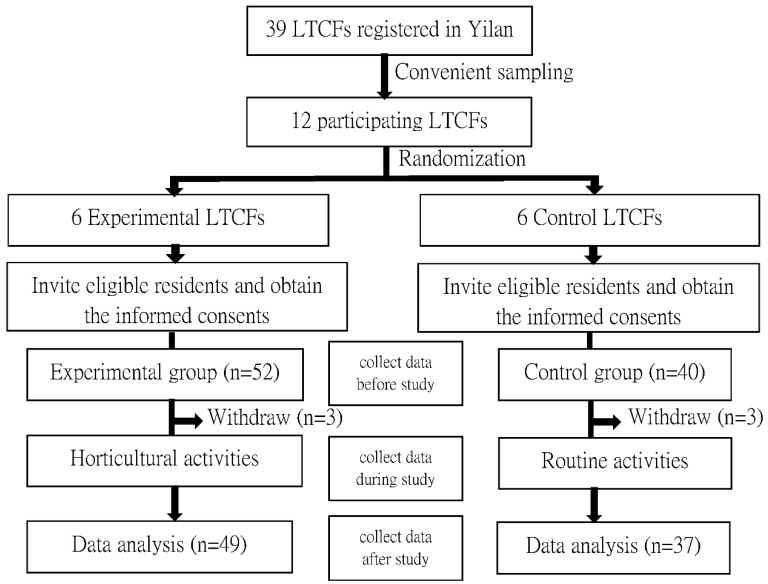
Flowchart of participant recruitment and program implementation.

**Table 1 ijerph-19-05412-t001:** Baseline characteristics of participants (*n* = 86).

Variables	Total (*n* = 86)	Experimental Group (*n* = 49)	Control Group (*n* = 37)	*p* Value
Gender, *n* (%)			0.501	
Male	31 (36.05)	16 (34.04)	15 (40.54)	
Female	55 (63.95)	33 (65.96)	22 (59.46)	
Age, year, mean (SD)	81.76 (8.47)	82.22 (7.47)	81.14 (9.72)	0.573
Marital Status, *n* (%)			0.667	
Unmarried	6 (6.97)	2 (4.08)	4 (10.81)	
Married	18 (20.93)	10 (20.08)	8 (21.62)	
Divorced or widowed	62 (72.10)	37 (75.50)	25 (67.57)	
Education level, *n* (%)			0.297	
Illiterate	43 (50.00)	23 (46.93)	20 (54.05)	
Elementary school	29 (33.72)	19 (38.77)	10 (27.03)	
High school and above	14 (16.28)	7 (14.30)	7 (18.92)	
Religion, *n* (%)			0.143	
None	13 (16.21)	7 (15.11)	6 (14.28)	
Yes	73 (84.79)	42 (84.90)	31 (85.72)	
Multi-morbidity, *n* (%)			0.101	
1 chronic disease	17 (19.76)	13 (26.53)	4 (10.81)	
≥2 chronic diseases	69 (80.24)	36 (74.47)	33 (89.19)	
Activities of Daily Living (ADL_S_), *n* (%)			0.940	
21–40	38 (44.18)	21 (42.85)	17 (45.94)	
41–60	32 (37.20)	19 (38.77)	13 (35.13)	
61–100	16 (18.63)	9 (18.38)	7 (18.93)	
Instrumental Activities of Daily Living (IADL_S_), *n* (%)			0.377	
1 item	34 (39.53)	16 (32.65)	18 (48.64)	
2 items	20 (23.25)	13 (26.53)	7 (18.91)	
3 items	14 (16.27)	10 (20.40)	4 (10.81)	
≥4 items	18 (20.95)	10 (20.42)	8 (21.64)	
Mini-Mental Status Examination (MMSE), *n* (%)			0.379	
<24	13 (5.11)	9 (18.36)	4 (10.81)	
≥24	73 (84.89)	40 (81.64)	33 (89.19)	
Geriatric Depression Scale (GDS), *n* (%)			0.690	
<5	68 (79.07)	38 (77.55)	30 (81.08)	
≥5	18 (20.93)	11 (22.45)	7 (18.92)	
Room type, *n* (%)			0.755	
Single room	12 (13.95)	6 (12.24)	6 (16.21)	
2–6 persons/room	74 (86.05)	43 (87.76)	31 (83.79)	
Presence of natural window view, *n* (%)			0.172	
Yes	34 (39.53)	22 (44.90)	12 (32.43)	
No	52 (60.47)	27 (55.10)	25 (67.67)	
Presence of outdoor public space, *n* (%)			0.204	
Yes	21 (24.41)	9 (18.36)	12 (32.43)	
No	65 (75.59)	40 (81.64)	25 (67.57)	
Number of leisure activities in LTCF per week, *n* (%)			0.547	
1–2 per week	63 (73.26)	37 (75.51)	26 (70.27)	
3–4 per week	23 (26.74)	12 (24.49)	11 (29.73)	
Number of family visits per week, *n* (%)			0.720	
1 per week	44 (89.79)	25 (51.02)	19 (51.35)	
2 per week	13 (15.11)	6 (12.24)	7 (18.91)	
3 per week	18 (20.93)	12 (24.48)	6 (16.21)	
≥4 per week	11 (12.79)	6 (12.26)	5 (13.53)	
Number of LTCF staff	15.53 (6.78)	19.55 (3.51)	10.22 (6.40)	0.083

SD: standard deviation, LTCF: long-term care facilities.

**Table 2 ijerph-19-05412-t002:** Sense of coherence score of the experimental group (*n* = 49) before and after intervention.

Time	SOC Score	Paired-Variable Difference	*t*	*p* Value
	Mean (SD)	Mean (SD)		
Baseline	50.45 (6.07)			
4th week	51.06 (6.28)	−0.61 (0.99)	−4.30	0.001
8th week	56.22 (7.19)	−5.77 (3.88)	−10.40	<0.001
12th week	56.37 (7.20)	−5.91(3.78)	−10.94	<0.001

SD: standard deviation.

**Table 3 ijerph-19-05412-t003:** Comparison of sense of coherence score between the experimental and control groups at various time points.

Groups	Baseline Mean ± SD ^1^	4th Week Mean ± SD	8th Week Mean ± SD	12th Week Mean ± SD
Experimental group	50.45 ± 6.07	51.06 ± 6.28	56.22 ± 7.19	56.37 ± 7.20
Control group	52.97 ± 6.00	53.03 ± 6.04	53.22 ± 5.96	53.22 ± 5.96
t	−1.91	−1.46	2.06	2.15
*p*	0.059	0.146	0.042	0.034

^1^ SD-standard deviation.

**Table 4 ijerph-19-05412-t004:** Generalized estimating equation analysis of sense of coherence score in the experimental and control groups before and after intervention.

95% Wald Confidence Interval						Hypothesis Test	
Parameter	B	Std. Error	Lower	Upper	Wald Chi-Square	df	*p* Value
intercept	52.97	0.97	51.06	54.88	2961.97	1	0.000
Experimental group vs. Control group (reference)	0.24	0.13	−0.05	0.044	3.71	1	0.071
Group × 12th week	5.96	6.22	4.74	7.18	91.78	1	<0.001
Group × 8th week	5.79	0.63	4.54	7.04	82.43	1	<0.001
Group × 4th week	0.53	0.15	0.23	0.84	11.93	1	0.001
scale	40.43						

## Data Availability

The data presented in this study are available on request from the corresponding author. The data are not publicly available due to privacy concern.

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
