# Peer review of "The Effects of Horticultural Therapy on Sense of Coherence among Residents of Long-Term Care Facilities: A Quasi Experimental Design"

_ijerph, 2022, doi:10.3390/ijerph19095412_

Round 1

Reviewer 1 Report

The manuscript contains an insufficient literature review. There are no references to the latest scientific publications on similar topics (also from 2021). An extended literature review can be placed in the Introduction chapter or a new Literature Review chapter can be added. 

Conclusions chapter needs to be corrected. Please try to refer to the experiences of researchers from other countries and try to compare them at least partially "To our knowledge, this is the first study reporting the horticultural therapy on sense of coherence among the institutionalized older adults in Taiwan".

Reviewer 2 Report

This study examined the effects of horticultural therapy of long-term care residents in sense of coherence by a quasi-experimental design. Horticultural therapy is often applied in long-term care facilities; however, the effects alternative therapies did not cumulative enough evidence in the past. This study would provide solid evidence to support this kinds of therapy in long-term care. Here are some comments or suggestions for the authors.

  1. Introduction: The theoretical explanation or the mechanism of horticultural therapy needs to be addressed. The authors mentioned salutogenic theory and ecological model but not explained in the Introduction.
  2. The paragraphs (line 89 – 104)“Horticultural therapy helps improve memory…. who live in 103 LTCFs still requires empirical investigation” describe the effects of horticultural therapy. These paragraphs are suggested to move to the Introduction section.
  3. The measurement of ADLs and IADLs are not described in the Method section 2.5. The references of ADLs, IADLs, MMSE, and GDS need to be cited. ‘
  4. The authors describe the descriptive analysis of the sample in the paragraph. If possible, to list in a table would better for readers.
  5. In the analysis of GEE, were the other variables (ADLs, IADLs, MMSE, etc.) also controlled in the model? If so, please state in the note. If not, I am wondering if these health variables added in the model still make the model robust.

Reviewer 3 Report

Very interesting research with significant  public health implications.

As it is , this draft can be published.

If the data availability permits it, i suggest to give more information and points of discussions on the following issues:

  • Are the SOC performances influence by sex, previous gardening experiences, socio economic categories?
  • What would be the potential of increased effectiveness of this therapy by use of incentives, support given to the gardening practice by supervisors, social /medical staff, provision of gardening materials, social recognition of results.....
  • Any economic implications? Cost effectiveness , cost benefits analysis?
  • Operational consideration: Constraints,advantages....to develop a sustained horticultural therapy
  • Any research suggestions to go beyond  SOC concept and link with other concepts

Reviewer 4 Report

The study investigated effects of horticultural therapy on sense of coherence (SOC), which is a new perspective. This study included enough number of participants, and horticultural activities were well designed. However, authors only focused on the changes of SOC score through the horticultural therapy. The current data is insufficient to discuss the relationship among horticultural therapy, SOC and the mental health of elderly people. In addition, there are some problems in methods and results as below.

Major Points

  • Methods and Results

Outcome measurement is few and insufficient. This study only focused on the changes of SOC score through the horticultural therapy. To meet the journal criteria, authors should have measured other changes associated with participants’ depression, anxiety or quality of life, and investigate the relationship between these factors and SOC scores.

  • Line 180, Data analysis

Two-way ANOVA (group and week) should be used to examine the changes of SOC sores. Repeated t-test is inappropriate.

Minor Points

  • Line 88-104

The descriptions should be included in introduction but not in methods.

  • Line 140

Are effects of all indoor horticultural activities same as outdoor ones? I recommend authors to describe the differences (or no difference) between indoor and outdoor therapy.

  • Line 201

Authors described that “There SOC scores for the 2 groups did not significantly differ (p < .0.59), indicating high homogeneity”. However, P value .059 means significant trend, the “high homogeneity” is not adequate.

  • Line 287

Why did the small-group horticultural activities improve SOC scores? I recommend authors to describe the relationship between horticultural thraphy and SOC in detail.

Round 2

Reviewer 1 Report

The chapter Conclusions still needs to be improved. Please add a few sentences that will emphasize the importance of the analyzes carried out in the manuscript.

Author Response

Manuscript ID: ijerph-1627087

Response to Reviewer 1 Comments (Round 2):

Point 1: The chapter Conclusions still needs to be improved. Please add a few sentences that will emphasize the importance of the analyzes carried out in the manuscript.

Response 1: Thank you for your great comment. As suggested, we have added a sentence in the conclusion to emphasize the importance of this analysis: “…, the results of the present study suggest that LTCF staff can incorporate horticultural activities into part of the LTCFs’ routine schedule to maintain or enhance the SOC level of their residents, which provides a reference for positive mental health improvement in the older people living in LTCFs.” (line 347-350, page 10)